# Enhancing Buprestidae monitoring in Europe: Trap catches increase with a fluorescent yellow colour but not with the presence of decoys

**Alexandre Kuhn**[1☯]*, **Gilles San Martin**[1☯], **Séverine Hasbroucq**[2], **Tim Beliën**[3], **Jochem Bonte**[4], **Christophe Bouget**[5], **Louis Hautier**[1], **Jon Sweeney**[6], **Jean-Claude Grégoire**[2]

1 Life Sciences Department, Crops and Forest Health Unit, Walloon Agricultural Research Centre, Gembloux, Belgium, 2 Spatial Ecology Laboratory (SpELL), CP 160/12, Université Libre de Bruxelles, Brussels, Belgium, 3 Zoology Department, Research Centre for Fruit Cultivation (pcfruit npo), Sint-Truiden, Belgium, 4 Flanders Research Institute for Agriculture, Fisheries and Food, Plant Sciences Unit, Merelbeke, Belgium, 5 INRAE, UR EFNO, Domaine des Barres, Nogent-sur-Vernisson, France, 6 Natural Resources Canada, Canadian Forest Service, Atlantic Forestry Centre, Fredericton, New Brunswick, Canada

☯ These authors contributed equally to this work.
* a.kuhn@cra.wallonie.be

**Data Availability Statement:** All the data and R scripts needed to reproduce our results are available in a public figshare repository: https://doi.

## Abstract

This study investigated the efficacy of various traps differing in colour (green or yellow), presence or absence of decoys (dead *Agrilus planipennis*) or design (commercial MULTz or multifunnel traps, and homemade *bottle-* or *fan-traps*) for monitoring European Buprestidae in deciduous forests and pear orchards. Over two years, we collected 2220 samples on a two-week basis from 382 traps across 46 sites in Belgium and France. None of the traps proved effective for monitoring *Agrilus sinuatus* in infested pear orchards (17 specimens captured in 2021, 0 in 2022). The decoys did not affect the catch rates whatever the trap model, colour, buprestid species or sex. The fluorescent yellow traps (MULTz and yellow *fan-traps*) tended to be more attractive than the green traps (green *fan-traps* and, to a lower extent, multifunnel green traps). Most *Agrilus* species showed similar patterns in mean trap catches, with the exception of *Agrilus biguttatus*, which had the largest catches in the green multifunnel traps. Finally, we observed a high variation in catch rates between localities: the site explained 64% of the catches variance, while the tree within the site and the type of trap explained only 6–8.5% each. In many sites, we captured very few specimens, despite the abundance of dying mature trees favourable to the development of Buprestidae. For the early detection of non-native Buprestidae, it therefore seems essential to maximise the number of monitoring sites. Due to their cost-effectiveness, lightweight design, and modularity, *fan-traps* emerged as promising tools for buprestid monitoring. The study's findings extend beyond European fauna, as a preliminary trial in Canada suggested that yellow *fan-traps* could also improve captures of non-European buprestid species and catch species of interest such as *Agrilus bilineatus* (a species on the EPPO A2 list of pests/pathogens recommended for regulation in the EU).

org/10.6084/m9.figshare.23982834 This DOI is currently inactive and will be made active should the manuscript be accepted for publication. A temporary private link provides access to this archive: https://figshare.com/s/e7fc272ba818a305fb53.

**Funding:** This research was part of the AGRITRAP project (RI 2020-A337) funded by the Belgian Federal Public Service (FPS) Health, Food Chain Safety and Environment. The funders had no role in study design, data collection and analysis, decision to publish, or preparation of the manuscript.

**Competing interests:** The authors have declared that no competing interests exist.

## Introduction

Wood-boring beetles (Coleoptera: Buprestidae, Cerambycidae and Scolytinae) include some of the most damaging forest pests. Furthermore, they are recognised as highly successful invaders and alien species are regularly intercepted [1–4]. The invasion potential of wood-borers is facilitated as they can easily travel unnoticed and protected in woody material. Proper management of both native and alien wood-boring beetles relies on efficient monitoring tools for their early detection and surveillance. Many species of Scolytinae and Cerambycidae are attracted by semiochemicals such as host volatiles and sex- or aggregation pheromones, and many synthetic lures are commercially available for use in survey traps [5–7]. In contrast, much less is known about the chemical ecology of jewel beetles. There is evidence of attraction to foliar and/or cortical volatiles emitted from host trees (especially stressed hosts) for a handful of buprestid species [e.g., *Agrilus planipennis* [8–12], *Agrilus anxius* [13], *Agrilus bilineatus* [14], *Agrilus biguttattus* [15], *Coraebus florentinus* [16], *Coraebus undatus* [17], and *Capnodis tenebrionis* [18]] and evidence of attraction to pheromones in *A. planipennis* [12,19–21], but the effect of semiochemicals on trap catch often depends on trap colour and where traps are deployed. For example, 3Z-lactone (emitted by female *A. planipennis*) significantly increased mean catches of *A. planipennis* [12,22] as well as the proportion of traps that detected at least one *A. planipennis* [19] so long as lures were placed on green traps co-baited with the green leaf volatile, Z-3-hexenol, and placed in the upper canopy of ash trees. So far, much work on the effects of visual stimuli on Buprestidae species has been done on the emerald ash borer (EAB), *A. planipennis* [23–26]. This species, native to Asia, was detected in 2002 in the USA [27]. Since then, it has spread to Canada and most eastern states of the USA and has become the most important pest of ashes (*Fraxinus* spp.) in North America. It was more recently introduced to European Russia and Ukraine and will likely expand its range westward to Europe [28]. Given its high economic impact, many studies have evaluated the performance of traps with different colours and shapes for monitoring or mass-trapping of EAB [29–34]. Other *Agrilus* species are also threatening the health of various tree species all over the world. In Europe, the two-spotted oak borer *A. biguttatus* is associated with oak decline [35–39], while damages caused by *A. viridis* were reported in beech forests from Central Europe [40,41]. In orchards, *A. sinuatus* is a well-known pest of pear trees [42,43]. These insects are usually considered as secondary pests, infesting weakened trees. However, more frequent drought and heatwave events due to climate change are expected to promote outbreaks by weakening the trees [44,45]. In this context, it is important to develop and validate efficient monitoring tools to assess the presence and the population level of native and non-native buprestid beetles, for European foresters and fruit growers. This is also relevant for non-European plant health services interested in early detection of alien pests.

Several studies have tested the efficiency of surveillance approaches developed for EAB (or inspired by them) toward European buprestid fauna [45–49]. Green and purple are the most widely used colours for buprestid trapping [7,32,33,49–51]. These colours indeed seem to attract European *Agrilus* species. Green sticky traps are efficient in capturing *A. angustulus*, *A. convexicollis*, *A. graminis*, *A. laticornis* and *A. obscuricollis*, while purple traps attract *A. biguttatus*, *A. sulcicollis* and *A. viridis* [39,46,52]. However, these colours were primarily used because of their attractiveness to EAB and other colours may be better suited to monitoring alternative species. Preliminary results indicated that yellow fluorescent colour may be very attractive to European *Agrilus* species [53].

Aside from trap colour, it has been suggested to use dead *Agrilus* specimens as decoys to enhance trap catches. Lelito et al. [23,54] observed that EAB males actively land on dead beetles of both sexes pinned on leaves and try to copulate ("paratrooper copulation"). Several studies

showed that the addition of such decoys (dead EAB or replicated models) on traps increase the number of *A. planipennis* captured [54–56]. This visually mediated direct approach has since been observed in other species, in response to conspecifics, but also to other *Agrilus* species [57,58]. For example, when presented with dead beetles pinned to foliage, males of the two-spotted oak borer landed more often on female *A. planipennis* than female conspecifics, and attempted copulation for longer periods [57]. Another study reports increased catches of European *Agrilus* species on sticky traps with pinned dead EAB compared to similar traps without decoys, confirming that decoys could improve trap attractiveness [46]. The vast majority of studies to date have investigated the decoy effect on trapping efficiency using sticky traps, which are usually small in size and generally placed directly on branches within living foliage [21,46,54–56,59] but see [60].

Currently, trapping of jewel beetles relies mostly on sticky (prism) traps or large multifunnel traps [30,31,33,45,47,51,52]. Sticky traps are notoriously difficult to handle, and saturation of the sticky surface by target and non-target insects is a typical limitation of such traps, reducing their efficiency over time [61]. Sticky traps are also normally discarded after one season of use whereas multifunnel traps can be reused for several years. In addition, trapped specimens can sometimes disappear from sticky traps if traps are not checked and specimens removed at regular intervals [62]. On the other hand, multifunnel traps (usually composed of 12 funnels) are heavy and bulky, hence harder to place in the tree canopy, and costly, limiting the number of monitoring sites. There is a need for effective, cheaper and lighter traps that are easy to transport and deploy in the upper tree canopy for monitoring jewel beetles [49].

In this study, we evaluated different trapping devices for the monitoring of native Buprestidae fauna in Belgium and France. In particular, we compared the efficiency of commonly used green multifunnel traps to alternative traps: commercial yellow fluorescent multifunnel traps called "MULTz" [53] and small, cheap and easy to deploy do-it-yourself (DIY) traps painted green or yellow. We also investigated the impact of decoys (dead EAB specimens) on trap catches. We focused our surveillance in suitable habitat for the known pests *Agrilus biguttatus*, *A. viridis* and *A. sinuatus* (i.e., oak trees, beech trees and pear orchards, respectively).

## Material and methods

### Monitoring design

We monitored buprestid beetles using different trapping designs over two years, 2021 and 2022. We placed the traps as high in the crown of the tree as possible, using a catapult or a very long telescopic arm (typically between 5 and 20 metres, depending on the height of the tree, and lower for the smaller *Pyrus* trees). We grouped the different trap models on the same tree and, where possible, on the same branch to make their catches as comparable as possible. We tried as much as possible to place traps on well spaced and sun-exposed branches (but this was not always on the south-facing side). Trap types were placed at random on each tree and the position of each trap was kept until the end of the field season None of the traps were baited with chemical lures. The traps were active from June to September and emptied every two weeks. The collecting pots of the traps were filled with 50% monopropylene glycol in water (Vital Concept, Loudéac, France). We brought the samples back to the lab and immediately transferred the collected insects to 80% ethanol and stored them at 4˚C. We then sorted the Buprestidae and morphologically identified each specimen under a stereomicroscope.

In 2021, we compared four trap designs in 13 Belgian sites located either in deciduous forest stands with oak (6 sites) or beech (2 sites) as dominant tree species, or in pear orchards (5 sites) (Table 1). We selected forest sites with senescent and/or dying trees characterised by dead branches and thin crowns, and orchards with a known history of *A. sinuatus* infestation.

**Table 1. Summary of the sampling design and number of captures.**

| Tree | Trap Type | NbIndiv | NbSites | NbTraps | NbSamples | NbDays |
|---|---|---|---|---|---|---|
| | | | Belgium—2021 | | | |
| Fagus | Bottle green | 0 | 2 | 6 | 30 | 107 |
| Fagus | Bottle green decoy | 0 | 2 | 6 | 30 | 107 |
| Fagus | Multifunnel green | 1 | 2 | 2 | 4 | 37 |
| Fagus | MULTz yellow | 1 | 2 | 5 | 20 | 89.2 |
| Pyrus | Bottle green | 5 | 5 | 15 | 135 | 101 |
| Pyrus | Bottle green decoy | 2 | 5 | 15 | 135 | 101 |
| Pyrus | Multifunnel green | 4 | 5 | 10 | 40 | 38 |
| Pyrus | MULTz yellow | 6 | 5 | 10 | 80 | 89.8 |
| Quercus | Bottle green | 48 | 6 | 18 | 118 | 100.5 |
| Quercus | Bottle green decoy | 63 | 6 | 18 | 119 | 101.3 |
| Quercus | Multifunnel green | 73 | 6 | 11 | 72 | 91.7 |
| Quercus | MULTz yellow | 605 | 6 | 12 | 77 | 98.2 |
| | | | Belgium—2022 | | | |
| Fagus | Fan-trap green | 1 | 3 | 6 | 47 | 122.8 |
| Fagus | Fan-trap green decoy | 1 | 3 | 6 | 48 | 125 |
| Fagus | Fan-trap green mixed | 0 | 1 | 1 | 3 | 89 |
| Fagus | Fan-trap yellow | 4 | 3 | 6 | 36 | 90 |
| Fagus | Fan-trap yellow decoy | 2 | 3 | 6 | 36 | 90 |
| Fagus | Fan-trap yellow mixed | 0 | 1 | 1 | 3 | 89 |
| Fagus | Multifunnel green | 24 | 4 | 7 | 51 | 122.2 |
| Fagus | MULTz yellow | 1 | 4 | 6 | 41 | 118 |
| Populus | Fan-trap green mixed | 0 | 1 | 2 | 6 | 89 |
| Populus | Fan-trap yellow mixed | 5 | 1 | 2 | 6 | 89 |
| Populus | Multifunnel green | 118 | 9 | 11 | 28 | 55.1 |
| Populus | MULTz yellow | 26 | 7 | 9 | 20 | 48.8 |
| Pyrus | Fan-trap green | 0 | 4 | 9 | 45 | 84 |
| Pyrus | Fan-trap green decoy | 0 | 4 | 9 | 45 | 84 |
| Pyrus | Fan-trap yellow | 0 | 4 | 9 | 45 | 84 |
| Pyrus | Fan-trap yellow decoy | 0 | 4 | 9 | 45 | 84 |
| Pyrus | Multifunnel green | 0 | 4 | 9 | 45 | 84 |
| Pyrus | MULTz yellow | 0 | 3 | 6 | 30 | 84 |
| Quercus | Fan-trap green | 133 | 7 | 14 | 109 | 113.5 |
| Quercus | Fan-trap green decoy | 114 | 7 | 14 | 110 | 114.4 |
| Quercus | Fan-trap green mixed | 4 | 2 | 4 | 12 | 90.5 |
| Quercus | Fan-trap yellow | 594 | 7 | 14 | 103 | 105.2 |
| Quercus | Fan-trap yellow decoy | 669 | 7 | 14 | 102 | 104.2 |
| Quercus | Fan-trap yellow mixed | 14 | 2 | 4 | 12 | 90.5 |
| Quercus | Multifunnel green | 554 | 10 | 20 | 131 | 114.9 |
| Quercus | MULTz yellow | 1401 | 10 | 20 | 132 | 115.7 |
| | | | France—2022 | | | |
| Quercus | Fan-trap green | 12 | 6 | 6 | 12 | 44 |
| Quercus | Fan-trap green decoy | 3 | 6 | 6 | 12 | 44 |
| Quercus | Fan-trap yellow | 56 | 6 | 6 | 12 | 44 |
| Quercus | Fan-trap yellow decoy | 32 | 6 | 6 | 12 | 44 |
| Quercus | Multifunnel green | 72 | 6 | 6 | 11 | 53.2 |

(*Continued*)

**Table 1.** (Continued)

| Tree | Trap Type | NbIndiv | NbSites | NbTraps | NbSamples | NbDays |
|---|---|---|---|---|---|---|
| Quercus | MULTz yellow | 166 | 6 | 6 | 10 | 48.3 |

Tree = genus of the tree in which the traps were hung. NbIndiv = total number of Buprestidae individuals captured, NbSites = number of sites, NbTraps = Number of Traps, NbSamples = number of samples, NbDays = average number of days each trap was active on the field. NB: The "Fan-traps mixed" concerns the few traps in which the samples with and without decoys have been pooled on the field.

The four trap designs for 2021 were: 1) multifunnel P218-Trap–green with 12 funnels (hereafter referred to as "multifunnel green"; Chemtica Internacional, Santo Domingo, Costa Rica); 2) yellow MULTz 4-funnel traps (hereafter referred to as "MULTz yellow"; Csalomon, Budapest, Hungary); 3) home-made green "*bottle-traps*" (hereafter referred to as "green *bottle-traps*") without decoys; and 4) green *bottle-traps* with one dead *A. planipennis* adult decoy glued in the middle of the interception zone of the trap. The *bottle-traps* are made from clear plastic and measure 26 cm high by 19 cm wide. We attached them by back-to-back pairs with a green plate (RAL 6038) in between (see Fig 1) and sprayed the interception zone with PTFE (Polytetrafluoroethylene) (WD40 for dry surfaces, Budd Lake, NJ, USA). In the forest sites, we hung the traps as high as possible in the crown of three trees per site (at least 50 m apart), each with all four trap types (multifunnel green, MULTz yellow, *bottle-trap* with decoy and *bottle-trap* without decoy) except the multifunnel green and MULTz yellow, which were only installed on two of the three trees (Fig 1A). Thus, treatments were replicated in a (incomplete) randomized block design, with individual trees acting as blocks nested inside sites, to control for variation in jewel beetle density among trees and sites. In pear orchards, we used the same design but we hung traps on adjacent trees rather than on the same tree because commercial pear trees are much smaller than forest trees.

In 2022, we compared six trap designs in 23 sites located in deciduous forest stands with oak (n = 15) or beech (n = 4) as dominant tree species or in pear orchards (n = 4) (Table 1). All sites were located in Belgium, except six of the oak sites that were located in France with a more restricted sampling period from June to the end of July (Table 1). We also placed some trap treatments in poplar stands to test their efficacy at detecting *Agrilus* species living on this

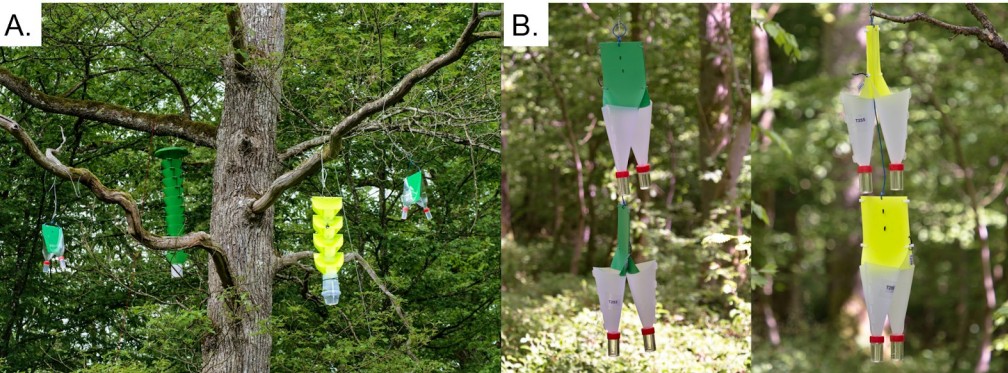

**Fig 1. Trapping devices used in this study. A**. Set-up used in 2021, from left to right: Green *bottle-trap* with decoy, multifunnel green, MULTz yellow, green *bottle-trap* without decoy hung in an oak tree. **B**. The *fan-traps* used in 2022 (replacing the *bottle-traps*). Left: Green fan-traps with (above) and without (below) decoys; right: Yellow *fan-traps* without (above) and with (below) decoys.

tree species (e.g. *Agrilus ater*, *A. pratensis*) but the design was irregular from site to site and these samples were excluded from most of the analyses (Table 1). For trapping, we re-used the multifunnel green and the MULTz yellow, but we replaced the green *bottle-traps* by recently developed DIY traps (hereafter referred to as "*fan-traps*"; [63]). The *fan-traps* are made from opaque polypropylene plastic sheets and measure 40 cm high by 15.5 cm wide. We painted the *fan-traps* either green (RAL 6038) or fluorescent yellow in order to test whether the success of the MULTz yellow in 2021 (see Results) was linked to its colour (see below for colour characterisation). The yellow paint (yellow fluor spray layer—Motip article number 04022) was applied over a first layer of plastic primer and a second layer of matt white (all products from Motip, Wolvega, The Netherlands). As done for the *bottle-traps*, we attached the *fan-traps* back-to-back by pairs of similar colour (Fig 1B) and sprayed the interception zone with PTFE. We tested the effect of decoys on the efficacy of *fan-traps* only, by glueing two dead EAB specimens (one placed directly above the other in the middle of the interception zone) on half of the green and yellow *fan-traps* (Fig 1B). We replicated the treatments in randomized blocks as in 2021, placing all six trap types (Multifunnel green, MULTz yellow, green *fan-trap* with decoys, green *fan-trap* without decoy, yellow *fan-trap* with decoys, yellow *fan-trap* without decoy) in each of two trees per site. We arranged pairs of *fan-traps* of the same colour (with and without decoys) directly adjacent to each other on a vertical axis, alternating traps with decoys in the top *vs*. bottom position to avoid potential bias (Fig 1B).

NB: There were some differences between the theoretical design presented here and the field reality in terms of exact number of visits, number of traps, number of trapping days, etc. due to practical constraints and problems in the field (e.g. trap destroyed or detached from trees, availability of some trap models, etc.). The actual design is summarised in Table 1 and in the S1 File. These discrepancies had no effect on the general conclusions of our analyses (see S1 File section 3.5 for details).

We characterised the colour of each trap using an ASD FieldSpec4 spectrometer (Malvern Panalytical, Malvern, United Kingdom). Briefly, we measured the reflectance spectra in the 350–2500 nm spectral range. For each trap, we took five reflectance measurements at different locations and averaged them to get the reflectance spectrum (see S1 File, section 6 for detailed results). For the visible light spectrum (380–700 nm), the green traps (i.e., *bottle-traps*, *fan-traps* green and multifunnel) had very similar spectra with a single peak of reflectance at 515, 515 and 528 nm, respectively. The yellow traps (i.e., MULTz and *fan-traps* yellow) had a main peak of reflectance at 545 and 527 nm, respectively. In addition, the MULTz had a small peak of reflectance at 404 nm, not present for the *fan-traps* yellow. Although the wavelength of the main absorbance peak is close in green and yellow traps, the reflectance spectra are quite different; for example, the reflectance remains high above 600 nm for the yellow traps compared to the green ones. (see S1 File, section 6).

## Data analysis

The statistical analyses and graphs were performed with the R programming language [64]. All the data and R scripts needed to reproduce our results are available in a public figshare repository: https://doi.org/10.6084/m9.figshare.23982834.

For all the models, we used residual plots to check the conditions of application of the model (mostly: linearity, variance and distribution of the residuals, outliers) and to guide the choice of transformations [65,66] (see S1 File).

Because we captured almost no buprestid beetles in pear orchards (see Results), these data were excluded from the analyses. We also excluded from analysis any replicate blocks (i.e., trees) in which none of the traps captured a single specimen of the particular response variable

being analyzed (for example if the response of the model was the total catches of *A. biguttatus*, we removed trees in which that species was not captured).

To standardise the sampling effort, we divided the total number of Buprestidae captured by the number of days each trap remained active. We then multiplied this number by 90 to obtain a corrected number of Buprestidae caught in each trap over a 90-day period (the traps remained active for 92.2 days on average). This value is hereafter referred to as "catch rate".

For most analyses, we used linear mixed models, with $\log(x+1)$ transformed number of catches per 90 days as response and the site and tree in which the traps were hung as random effects. The fixed effects and subset of data used varied depending on the hypotheses we wanted to test. For the tests (Wald F tests), the degrees of freedom were computed with the Kenward-Roger method (package 'car', [67]). NB: we also tested Generalized Linear Mixed Models (GLMM) with Poisson or Negative Binomial distribution (and log number of catching days as an offset) but these models frequently caused problems of model convergence while the Gaussian models with log transformed response fitted fine and had usually good residual plots (see S1 File). When it was possible to fit both Gaussian and Poisson GLMMs, the conclusions were very similar and our conclusions were globally very robust to the analytical choices or the subset of data used (see S1 File section 3.5 for details).

**Impact of decoy on trap catches.** We first investigated whether the addition of an EAB decoy had any effect on catches on a subset of the data including only the types of traps with or without decoys (i.e. *fan-traps* and *bottle-traps*). Our first model included the type of trap, the presence of a decoy and their interaction as fixed effects. To determine whether the effect of a decoy is species-specific, we used the same model but we added the Buprestidae species as a qualitative explanatory variable and the decoy x species interaction. We performed this analysis on a subset of data including only seven species with enough captures (*Agrilus angustulus*, *A. biguttatus*, *A. graminis*, *A. laticornis*, *A. obscuricollis*, *A. olivicolor* and *A. sulcicollis*).

Finally, we tested whether the effect of a decoy was sex-specific (i.e. was more attractive to males or to females). Again, we made the analysis on a different subset of data because we had no sex information for some sites. We used the initial model including only the sex, presence of decoy and their interaction as fixed explanatory variables to test the effect of the sex, whatever the species captured. Then, we used the initial model but included species, sex, presence of decoy and all interactions as fixed explanatory variables to test whether attraction to the decoy differs between sexes for some species but not for others.

**Impact of trap type on catches.** Since we found no effect of the presence of a decoy on trap captures (see Results), we pooled the captures from the pairs of *fan-traps* or *bottle-traps* with and without decoys hung in the same tree.

We used trap type (multifunnel green, MULTz yellow, green *bottle-trap*, green *fan-trap* or yellow *fan-trap*), species of the trap-bearing tree and year of monitoring as qualitative explanatory variables. In some of the monitoring sites, we had almost no buprestid captured (see Results). Such data tend to reduce the differences between trap types (as they all have null catches) because of the absence of buprestid beetles in the site rather than because of a similar attractivity of the traps. To evaluate the robustness of our results, we repeated the previous analysis on a subset of data including only trees with a total number of catches (from all traps hung on this tree) greater than or equal to five.

To test the effect of sex and buprestid species, we had a similar approach as the one explained above to compare the traps with and without decoys: a model with trap type, buprestid species and their interaction (on a subset of the most common *Agrilus* species), another model with sex, trap type and their interaction (for a subset of the sites for which sex level identification was available) and a final one with trap type, buprestid species and sex. To simplify these already rather complex models, we did not include the year (which seemed to have

limited effect on the results based on the model using the total number of individuals as response) nor the tree species because most of the subsets concerned only data from oak tree forests.

Because we found a significant trap type x species interaction (see Results), we also fitted separate mixed models for each species and performed all pairwise comparisons between trap types with p-value corrections for multiple testing (default single-step method of the "mult-comp" package [68]).

To determine the proportion of variance explained by the trap type *vs.* the position of the traps (i.e. the site and the trap-bearing tree), we fitted a fully random effects model with trap type, site and tree as random effects [69].

Finally, we compared the effect of trap type on the number of species captured and on the probability of detecting each species. This last model was a GLMM with a binomial residual distribution, the presence/absence of each species per trap as response, the type of trap, species and their interaction as fixed effects and the usual sites and tree factors as random effects.

## Results

Overall, we captured 4814 Buprestidae specimens, including 4669 of the genus *Agrilus*. We collected 2220 samples from 382 traps spread over 46 sites in France and Belgium (Table 1). On average, each trap remained active for 92.2 days. Most captures took place in June to July and declined in August. The number of catches varied very much between sites (range of the number of Buprestidae captured per trap per 90 days: 0 to 400, median = 0, average = 11.6). Over the two years of study, the traps captured 23 different species (20 in Belgium and 12 in France, where the sampling effort was lower) (Fig 2). The seven most abundant species were *Agrilus sulcicollis*, *A.laticornis*, *A.angustulus*, *A.olivicolor*, *A.graminis*, *A.obscuricollis*, and *A.biguttatus*, accounting for 92.7% of the total catches (Fig 2). In Belgium, the most abundant species was *A. sulcicollis* with 1925 specimens captured (a catch rate of 5.2 specimens/trap/90 days). This species occurred much less frequently in France (ranking 6th with a total of 19 specimens captured, i.e. 1.03 specimens/trap/90 days). In France, *A. laticornis* was the most frequent species, with 123 specimens (5.2 specimens/trap/90 days). The highest number of species was found in oak stands (21 species), while we found only *A. sinuatus* in pear orchards. The catches were particularly poor in pear orchards: we captured only 17 specimens of *A. sinuatus* in 2021 and none in 2022 despite a confirmed presence of the species in all orchards as determined by branch-beating. For oak stands, the fauna from Belgium and France was quite similar, with a few species found only in traps from France (*A. hastulifer*, *A. curtulus*, *Meliboeus fulgidicollis*) and higher relative frequencies for some species like *A. obscuricollis* and *C. undatus*. In poplar sites, we also captured two species strictly associated with Salicaceae: *A. ater* (14 specimens) and *A. pratensis* (120 specimens).

### Impact of EAB decoys on trap catches

We found no significant trap x decoy interaction ($F_{2, 82.5} = 0.39$, p = 0.68) nor any main decoy effect ($F_{1, 82.5} = 1.37$, p = 0.24) on the number of buprestids captured (Fig 3). The effect of trap type was, however, highly significant ($F_{2, 77.8} = 26.42$, p < 0.001) and is further investigated below. These results suggest that the presence of an EAB decoy on *bottle-traps* or *fan-traps* does not increase buprestid catch rates. The results were similar when investigating whether the decoy effect could be species-specific, using the seven most frequent species: the decoy x species and the decoy x trap type x species interactions were not significant ($F_{6, 383.1} = 0.45$, p = 0.84 and $F_{12, 383.1} = 0.23$, p = 1, respectively). In contrast, the effects of buprestid species ($F_{6, 398.5} = 30.49$, p < 0.001) and trap type ($F_{2, 69.6} = 52.31$, p < 0.001) were highly significant.

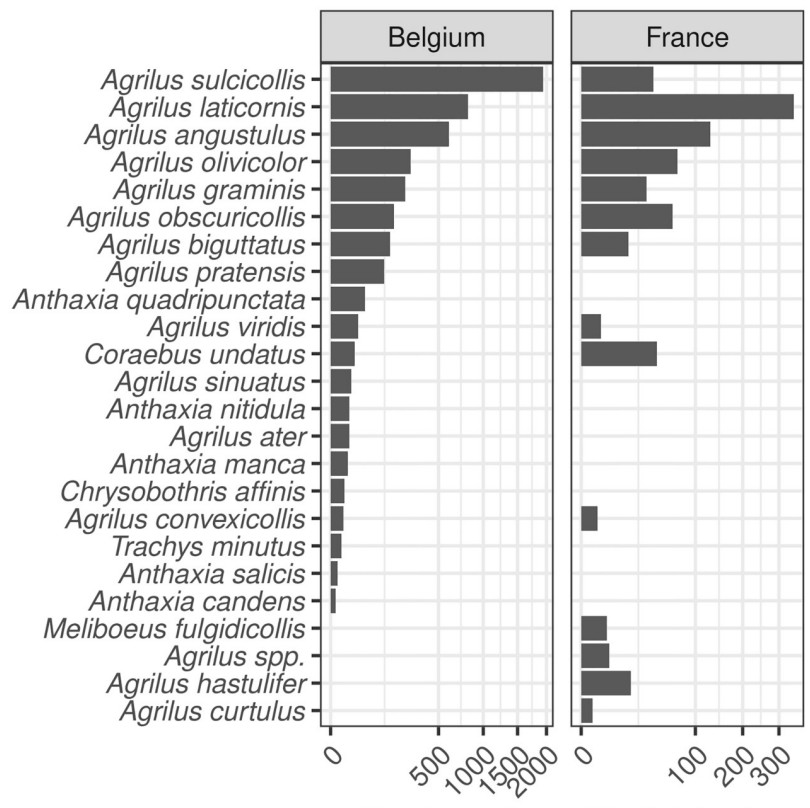

**Fig 2. Total number of individuals of each species captured in France and Belgium over the two years of this study and for all tree species (*Fagus sylvatica*, *Populus* spp., *Pyrus communis* and *Quercus* spp.).** NB: The sampling effort was much lower in France. *Agrilus spp.* designate specimens that were too damaged to be identified. The catches for each tree species can be found in the supplements (S1 File section 2.3).

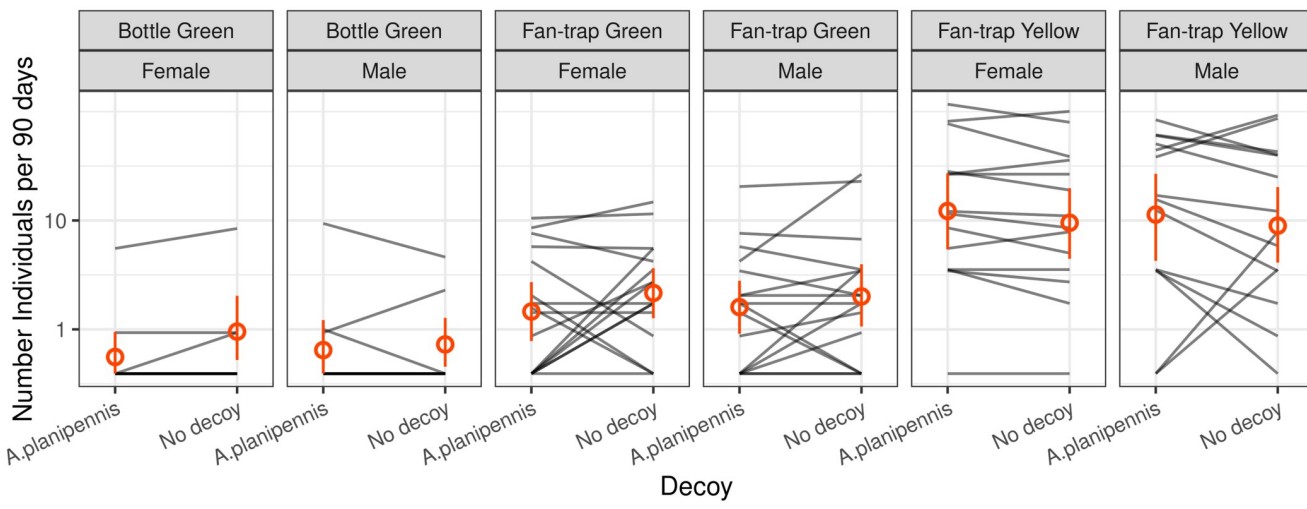

**Fig 3. Comparison of the number of Buprestidae captured by different trap types and for each sex with or without a glued dead *Agrilus planipennis* as a decoy.** The grey lines link similar trap types hung in the same tree and hence directly comparable. The red dots represent the averages, and the bars represent the 95% bootstrap confidence intervals. The presence of decoys has no effect on the number of captures. The supplements, S1 File section 3.1, also provide a detailed analysis, species by species.

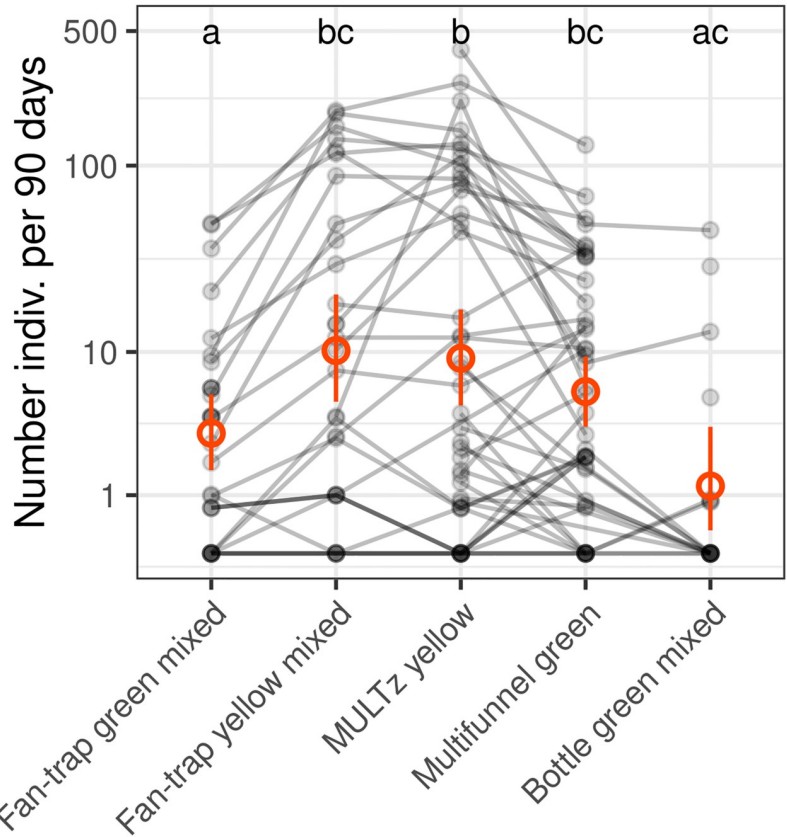

**Fig 4. Comparison of the number of Buprestidae captured by different trap types.** We pooled the catches from the *fan-traps* and *bottle-traps* with and without decoys hanging from the same tree. The grey lines link traps hung in the same tree. The red dots represent the averages and the bars represent the 95% bootstrap confidence intervals. The letters summarise the *post-hoc* tests for all pairwise comparisons: Different letters indicate a significant difference (after p-value correction for multiple testing). The yellow coloured traps tended to capture more than the green *fan-traps* and to a lower extent than the multifunnel green traps, but the variation between sites is very large. Note that the comparison between *bottle-traps* and *fan-traps* is more uncertain because these traps were not deployed in the same sites nor in the same year.

There was also a significant trap type x species interaction ($F_{12, 401.3} = 3.21$, $p < 0.001$), indicating that the impact of the trap type on catch rate was not the same for all species. Finally, we investigated whether the decoy effect was sex-specific, i) considering the species separately, or ii) pooling species. Both analyses led to the same results, revealing no significant effects or interactions for the sex nor the decoy variables (Fig 3; see S1 File section 3.1.3 for details).

## Impact of trap type on catches

We first investigated the effect of the trap type on the catch rate, including as covariates the tree species (*Fagus sylvatica* or *Quercus* spp.) in which the traps were hung and the year of experiment. Our results show a highly significant effect of trap type ($F_{4, 110.3} = 12.01$, $p < 0.001$), and tree species ($F_{1, 23.1} = 5.19$, $p = 0.03$) on catch rate, but no effect of the year of experiment ($F_{4, 55} = 0.82$, $p = 0.37$). Fig 4 summarises the pairwise differences between trap types. The green *fan-traps* captured significantly fewer specimens than the yellow *fan-traps*, MULTz yellow and multifunnel green models. The yellow *fan-traps* captured the highest average numbers, closely followed by the MULTz yellow and the multifunnel green traps, but the

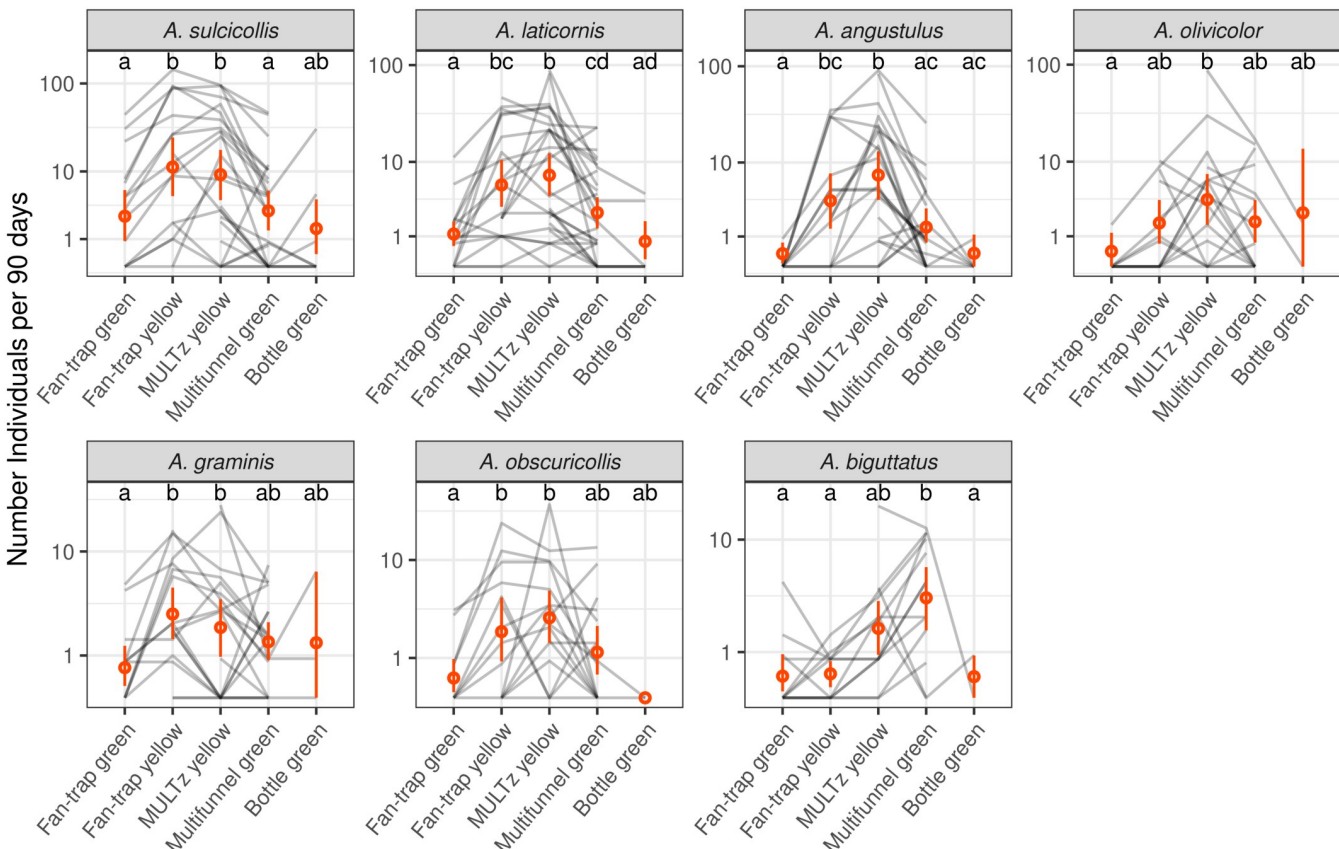

**Fig 5. Comparison of the number of *Agrilus* specimens captured by different trap types for the seven most common species accounting for 92.7% of the catches.** The grey lines link traps of different types hung in the same tree. The red dots represent the averages and the bars represent the 95% bootstrap confidence intervals. The letters summarise the *post-hoc* tests for all pairwise comparisons: Different letters indicate a significant difference (after p-value correction for multiple testing). The general trends are similar between species except for *A.biguttatus*. Note that the comparison between *bottle-traps* and *fan-traps* is more uncertain because these traps were not deployed in the same sites nor in the same year.

differences between these three trap types were not statistically significant. The green *bottle-traps* captured very few individuals, but differences generally remain non-statistically significant, probably due to the limited number of traps and high variability between sites (Table 1). Regarding tree species, traps hung in oaks captured significantly more buprestids than traps hung in beech trees. The sites in which the traps were placed had a considerable impact on the variation in the number of captures. Differences between sites explained 64% of the total variation. The tree in which the traps were hung within the site accounted for 6% of the variance, which is only slightly less than the variance explained by the trap type (8.5%).

We then investigated whether the efficacy of the different traps varied according to species of jewel beetle. The model indeed shows a significant trap type x species interaction ($F_{24,\ 468.2}$ = 1.69, p < 0.02), which indicates that the trend in catch rate among trap types was not the same for all species. Looking at each species separately, we observed a similar trend in all species except *A. biguttatus* (Fig 5). Even though the differences were not always significant, catch rates of most species tended to be higher in yellow traps (yellow *fan-traps* and MULTz yellow) than in green traps. However, catch rates of *A. biguttatus*, were significantly higher in the multifunnel green traps compared to the green and yellow *fan-traps* and to the green *bottle-traps*

(note that the *bottle-traps* caught a single specimen of *A. biguttatus*), while the MULTz yellow showed intermediate captures, not significantly different from the other trap types.

We tested whether catch rates among trap types were affected by the sex of the specimens. When pooling captures from all species, we found no evidence that the beetle sex influenced the mean catch rate ($F_{1, 130.5} = 1.33$, p = 0.25), nor the captures according to trap types ($F_{4, 130.5} = 0.77$, p = 0.55). When we included the species as an explanatory variable, we found no significant main effect of the sex on catch rate, but did find a significant species x sex interaction ($F_{6, 814.4} = 3.06$, p = 0.0062), suggesting differences in captures between males and females depending on the *Agrilus* species (but independent of the trap type). However, when comparing male *vs*. female catches for all combinations of species and trap types (with p-value correction for multiple testing), none were significantly different (S1 File section 3.2.5).

The number of species per trap showed a strong positive log-linear correlation with the number of individuals caught (Pearson r = 0.94, see S1 File, section 3.3). Consequently, the trap types that caught more individuals tended to catch also more species: green *fan-traps* caught an average of 2.1 species per trap, i.e. significantly less than yellow *fan-traps* (3.6), MULTz yellow (3.5) and multifunnel green traps (2.9). The probability of detecting each species (presence/absence data) was also lower for the green *fan-traps* compared to the yellow *fan-traps* and the MULTz yellow traps, with the multifunnel green traps in between. These trends were similar for the different species (no species x trap type interaction, Chisq = 18.17, df = 18, p = 0.44, see S1 File, section 3.4 for details).

## Discussion

We investigated the efficacy of different trap designs for monitoring the Buprestidae fauna associated with oaks, beeches and pear orchards in Belgium and France. The number of buprestid specimens captured per trap in a 90-day period varied greatly among sites and to a lesser extent between individual trees within sites and between trap types. Adding a dead EAB as a decoy did not increase catch rates on either *bottle-traps* or *fan-traps*. Traps with a yellow fluorescent colour tended to perform significantly better than green traps for all of the most frequently captured species, with the exception of *A. biguttatus* which was captured in greater numbers with green multifunnel traps. None of our trapping devices were effective at capturing *Agrilus sinuatus* in pear orchards. Finally, trap type did not influence the sex ratio of the beetles captured, indicating that all trap types performed similarly for both sexes.

The most common species in our dataset, *A. sulcicollis*, was surprisingly captured much less frequently in France than in Belgium. This could be due at least partially to a sampling artefact. Indeed, sampling in 2022 started around 31st May in France compared to 12th May in Belgium. Since *A. sulcicollis* tends to be active earlier in the season (S1 File section 5.2) we may have missed the peak of its activity in France. Other species had a higher relative frequency in France, particularly *C. undatus* and *A. obscuricollis*. *Coraebus undatus* was previously known only from two old records (<1945) in Belgium and *A. obscuricollis* and *A. graminis* had not been previously reported in Belgium [70,71]. We are most likely witnessing a Northward expansion of several species related to global warming, as already reported for various Buprestidae [37,72]. The natural expansion of *C. undatus* might be of particular concern as this species is generally considered a pest of oak trees in France and Southern Europe [37,73].

Globally, our traps performed well: we captured 4814 Buprestidae in total and up to 400 specimens in a single trap. However, we observed high variability between sites, probably due to differences in local population densities. The most favourable sites were all located in oak forests (Table 1). Yet, we captured only a few specimens (<20) at several oak sites, despite moderate to high levels of oak dieback. This indicates that, besides declining trees, additional

factors (e.g. climatic) are necessary for Buprestidae to establish or that the dieback was already too advanced and no longer provided suitable brood hosts for *Agrilus* spp. The number of captured Buprestidae also varied between trees within monitoring sites. According to our results, the choice of the individual tree in which the traps were placed was as important as the trap type in explaining the variability in trap captures. In line with this, previous studies reported a significant effect of trap position and environment on catches (height, exposition, size of log piles stacked nearby) [46,74,75]. Consequently, the choice of the trap-bearing trees appears crucial when setting up monitoring surveys.

In pear orchards, we only captured 17 specimens of *A. sinuatus* in 2021 and no specimens at all in 2022. Yet, their presence in the orchards was attested by branch-beating tray monitoring. We can thus reasonably conclude that none of the trapping systems tested in this study were effective for monitoring this species. The yellow and green colours might be less attractive for *A. sinuatus* than for other species. *A. sinuatus* has itself a characteristic bronze colour which differs from many species living on oak trees and could maybe explain a lower attractivity to the trap colours used in this study. Furthermore, the conditions (exposure, contrast) to which the traps were subjected in the pear orchards and in the forest stands are very different and may have reduced the attractiveness of the traps in the orchards. It is also possible that yellow and green colours are indeed attractive but that most *A. sinuatus* attracted to the immediate trap vicinity fail to be captured. In this case, the use of small sticky traps (possibly with *A. sinuatus* decoys) may be more suitable (e.g. branch traps used in [21]). In all oak, beech and poplar sites, we hung the traps high in the canopy of mature trees. However, in the pear orchard sites, all trees are small and the traps were hung at eye level. This might also influence the behaviour of the beetles. Other trap designs and colours should therefore be tested in the future to facilitate the monitoring of this economically important species. It has been shown, for example, that egg-yolk yellow and light green sticky traps with Z-3-hexenol lure were attractive for the related species *Agrilus mali* [76]. Such combinations might therefore be a good option for future trials.

In our study, adding dead specimens of *Agrilus planipennis* as decoys did not affect trap catches, whatever the species considered, their sex or the trap colour (Fig 3). This result is surprising since several previous studies reported higher captures for several *Agrilus* species, including European ones, when adding such decoys [46,48,56]. However, our results agree with Santoiemma et al. [60] who recently reported no significant effects of Agrilus decoys on abundance or species richness of *Agrilus* spp.captured in non-sticky green intercept panel traps. That study found a positive effect of *A. bilineatus* decoys on catch of only 1 of 25 species tested, *Agrilus geminatus*, and found negative effects of *A. sulcicollis* and *A. laticornis* decoys on catch of *A. sulcicollis* and *A. hastulifer*, respectively.

Several non-exclusive hypotheses may explain this discrepancy. First, the attractivity of decoys seems to depend on a subtle combination of visual and olfactory cues, and the right combination can differ from one target species to another [48]. For example, in Lelito *et al.* [56], *Agrilus cyanescens* (a species also present in Europe) strongly reacted to *A. planipennis* decoys on blue sticky traps but not on yellow ones. Domingue *et al.* [46] detected a decoy effect only on green sticky traps, but not on four other types of traps with various colours. Thus, it might be possible that our combination of trap background colour, decoy and native species explains our negative results. Second, all experiments showing an effect of decoy relied on sticky traps, while we used only non-sticky interception traps. In addition, Domingue et al. [46] added a thin layer of glue on top of their decoys, while ours were simply glued to the traps. Some species of *Agrilus* seem to land directly on the decoys, while others land on the foliage and then engage in antennal contact with the decoys [48]. In our setting, a male landing directly on the decoy might be able to fly away. However, Lelito et al. [56] did not add glue on

their decoys. Nevertheless, they reported higher captures of *A. cyanescens* on traps with decoys even though they observed this species landing directly on these decoys. Third, positive response to decoy was mostly reported in association with small sticky traps placed in the branches, within the foliage [21,46,54–56,59]. This position may provide the necessary visual and olfactory stimuli to allow the decoys to work. In contrast, decoys added to stand-alone traps hung to branches may not elicit male mating behaviour due to inappropriate context. Finally, the decoy attractivity described in the literature for European species appears rather inconsistent. For example, in the only published European studies [46] a positive decoy effect was detected only on green sticky traps and only for *A. sulcicollis* or the sum of all species in one study [46] whereas there was a negative effect of decoys on catch of *A. sulcicollis* in the other study [60]. Further replication studies might be needed to clarify whether decoys are really improving captures for European Buprestidae and with which combination of visual (and chemical) cues.

We found important differences in captures between trap types (Fig 4). The MULTz yellow and yellow *fan-traps* captured the highest number of buprestid specimens, closely followed by the multifunnel green. The green *fan-traps* were clearly less efficient. The green *bottle-traps* caught significantly fewer specimens than MULTz yellow, and only a limited proportion of *bottle-traps* actually captured any specimens (17% of *bottle-traps* with >0 specimens). The *bottle-traps* were used only during the first year of the project, in a limited number of sites, most of the time with very low local population densities (Table 1). The comparison with the other trap types is therefore not straightforward. The two most efficient traps were yellow fluorescent, suggesting this colour is more attractive to most European *Agrilus* species than the green colour more commonly used in commercial traps. The superiority of the MULTz yellow over the multifunnel green to catch European species (*A. angustulus*, *A. graminis*, *A. laticornis*, *A. obscuricollis* and *A. olivicolor*) has been reported in a previous study [53]. However, the two multifunnel models compared by Imrei et al. [53] were rather different and the greater efficacy of the MULTz yellow could be related to its colour, shape, or a combination of both factors. In contrast, comparing the catches of the green and yellow *fan-traps* clearly shows the greater efficacy of the yellow fluorescent colour over the green one. The preference for yellow fluorescent colour does however not seem to be a universal rule. Although most species followed this trend, *A. biguttatus* was captured in higher numbers in multifunnel green traps (Fig 5). These higher catches appear to be related to the trap type as a whole and not just the green colour, as green fan-traps performed poorly for *A. biguttatus* and were no better than yellow *fan-traps* (Fig 5). A similar preference for multifunnel green over MULTz yellow has also been demonstrated for *A. planipennis* males [53]. *Agrilus planipennis* and *A. biguttatus* were shown to have similar reflectance patterns and males *A. biguttatus* are more responsive to *A. planipennis* decoys than to conspecific ones [57,77]. It is thus not surprising that they share a common preference for trap colour.

Aside from trap colour, the surface area of the traps could also influence the number of captures. Indeed, multifunnel traps are much larger than *bottle-* and *fan-traps* (even pooled pairs of *fan-traps* with and without decoys). One could hypothesise that yellow *fan-traps* of the same size would perform even better than MULTz yellow ones. When we correct the number of captures by the trap surface area, the yellow *fan-traps* become indeed the most efficient traps (see S1 File section 3.2.3). However by doing so we impose on our data a linear relationship between trap surface area and the number of catches. In order to formally test such a size effect, we would have needed similar traps of varying sizes. Moreover, beyond the total trapping surface, the trap structure is likely important and it may be more efficient to stack multiple smaller *fan-traps* rather than enlarging the interception panel. The *fan-traps* have been specifically designed to be cheap, light, small and easy to combine and set up in numerous

locations [63]. For monitoring purposes, placing cheap traps in many different locations is very likely to be more cost effective since we showed that trapping location is more important than trap type to explain the number of catches. In addition, these simple traps would make it easy to present multiple combinations of visual or chemical attractants, depending on the preferences of the target species.

Our results indicated a similar sex ratio in all trap types tested but differing slightly between species, with generally slightly more males for several species (but more females for *Agrilus graminis*). According to previous studies, sex-specific attractivity can be colour dependent. For *A. planipennis*, captures in green traps are balanced or skewed toward males, whereas purple traps are more attractive to females [30,34]. The same pattern is present in *A. sulcicollis* and *A. bilineatus* where females are also more attracted to purple traps while no preference is present in males [78]. In Europe, Rhainds et al. [52] showed that purple traps captured mostly female *A. viridis*, while green trap captures were biased toward males in *A. convexicollis*. It has been hypothesised that, for species developing in the trunk or large branches, such as *A. biguttatus* or *A. sulcicollis*, green colour could be more similar to foliage, attracting both sexes for feeding and mating, while darker colours (purple) could be more similar to trunk and branches reflectance, thus being more attractive to females looking for oviposition sites [30,51]. In our study, yellow fluorescent and green colours may thus mimic foliage and attract equally males and females.

According to our results, the *fan-traps* performed similarly or better than commercial multifunnel traps, but for a reduced cost: around 3 euros/trap (for a single unit, they are grouped by 2 or 4 in our study, please see the Material and Methods for details) for the fan-traps (including colour but not manpower; 59) against 45 euros/trap (including shipping fees) for the multifunnels (price based on Chemtica Internacional, Costa Rica, and Csalomon, Hungary, offers in 2021 for shipping to Belgium). Although *fan-traps* require some handling for assembly, they allow greater modularity (colour, disposition, initial size). Given their low cost and light design, they can be used to monitor more locations. As stated above, this strategy would most likely increase detection and captures given the importance of trapping location. In our study, the yellow-fluorescent *fan-traps* were particularly efficient at capturing *Agrilus* species associated with European oak, including *A. sulcicollis* which has recently established in North America and is under surveillance [78–80]. Unfortunately, multifunnel green traps seem more appropriate for *A. biguttatus* which is recognised as the most damaging species to oak trees in Europe [37]. When designing a surveillance programme, the cost and effectiveness of the monitoring equipment should be weighted according to the economic importance of the target species. On the other hand, we have shown that the number of species captured is strongly correlated to the total number of individuals per trap. So, for a general purpose surveillance program targeting a broad range of species, choosing traps which maximize the number of catches can be the best option if resources are limited.

It would also be interesting to test the efficacy of *fan-traps* on non-European buprestid fauna for monitoring exotic species in Europe. In line with this, a small preliminary trial was initiated in 2022 by sending green and yellow *fan-traps* to New Brunswick, Canada. These traps were deployed in a mature red oak, *Quercus rubra*, stand and hung in a tree together with a multifunnel green trap (for a total of 3 trees and 9 traps). The multifunnel green, but also the yellow *fan-traps* captured several specimens of different species, indicating that yellow *fan-traps* are also attractive to non-European buprestid fauna (see S1 File, section 4.5). As in Europe, the green *fan-traps* captured much fewer specimens but this should be confirmed with more replicates. All types of traps captured *A. bilineatus*, a species under surveillance in Europe (EPPO A2 list) and a quarantine pest in the UK.

## Conclusion

Our results indicated that, for monitoring European Buprestidae inhabiting deciduous forests, yellow fluorescent traps, namely the MULTz yellow and the yellow fluorescent *fan-traps*, captured the highest number of buprestid beetles, followed by the multifunnel green. Green *fan-traps* were less efficient, while the homemade green *bottle-traps* were highly variable, giving inconsistent results. Trapping efficacy was species-dependent and multifunnel green traps appeared more attractive to *A. biguttatus*, whereas other common species preferred yellow fluorescent traps. We also found that adding one or two dead *A. planipennis* specimens as decoys did not enhance trapping efficacy, with the traps tested in this study. On the other hand, our results confirm that the site and the tree in which traps were hung within the site strongly influenced trapping success.

Surprisingly, none of the trapping devices used in our study was effective in monitoring of the pear pest *A. sinuatus* and alternative designs will need to be tested for its surveillance in pear orchards.

Given their low cost, light weight and modularity, the *fan-traps* appear as an efficient and versatile trap design for monitoring buprestid beetles. In particular, the yellow fluorescent *fan-traps* used in this study are as effective as large, expensive commercial multifunnel traps. A small-scale trial carried out in Canada indicated that yellow fluorescent *fan-traps* could also be effective in capturing non-European fauna, including *A. bilineatus*, a species that is considered as potentially invasive if established in Europe. Their efficacy to catch invasive species such as *A. planipennis* remains to be tested.

## Supporting information

**S1 File. Detailed data and statistical analyses.**
(PDF)

## Acknowledgments

This research was part of the AGRITRAP project (RI 2020-A337) funded by the Belgian Federal Public Service (FPS) Health, Food Chain Safety and Environment. We warmly thank many people for their technical help in the field or for samples handling and identification: Cory Hughes and Chantelle Kostanowicz (Canadian Forest Service); Carl Moliard and Guilhem Parmain (INRAE Orléans, France); Emilio Caiti (ULB, Belgium); Olivier Vanhoutte (ILVO, Belgium); Peter Dewitte (PC Fruit, Belgium), Mallory Akhamlich, Quentin September, Marie Tomme, Cyril Vos and Anne-Michèle Warnier (CRA-W, Belgium). We thank Ben Slager (USDA, USA) for sending dead specimens of *Agrilus planipennis*. We also thank the Walloon Forest administration (DNF) and the Donation Royale for granting access to some of the monitoring sites. Finally, we thank Damien Vincke and Benoît Scaut (CRA-W, Belgium) for the spectrometer measurements.

## Author Contributions

**Conceptualization:** Alexandre Kuhn, Gilles San Martin, Tim Beliën, Jochem Bonte, Louis Hautier, Jean-Claude Grégoire.

**Data curation:** Alexandre Kuhn, Gilles San Martin, Séverine Hasbroucq, Jochem Bonte.

**Formal analysis:** Alexandre Kuhn, Gilles San Martin.

**Funding acquisition:** Gilles San Martin, Tim Beliën, Jochem Bonte, Louis Hautier, Jean-Claude Grégoire.

**Investigation:** Alexandre Kuhn, Gilles San Martin, Séverine Hasbroucq, Christophe Bouget, Jean-Claude Grégoire.

**Methodology:** Alexandre Kuhn, Gilles San Martin, Séverine Hasbroucq, Tim Beliën, Jochem Bonte, Jon Sweeney, Jean-Claude Grégoire.

**Project administration:** Alexandre Kuhn, Gilles San Martin, Tim Beliën, Jochem Bonte, Louis Hautier, Jean-Claude Grégoire.

**Resources:** Tim Beliën, Jochem Bonte, Christophe Bouget, Louis Hautier, Jon Sweeney, Jean-Claude Grégoire.

**Software:** Gilles San Martin.

**Supervision:** Jochem Bonte, Christophe Bouget, Louis Hautier, Jon Sweeney, Jean-Claude Grégoire.

**Visualization:** Alexandre Kuhn, Gilles San Martin.

**Writing – original draft:** Alexandre Kuhn, Gilles San Martin.

**Writing – review & editing:** Alexandre Kuhn, Gilles San Martin, Tim Beliën, Jochem Bonte, Christophe Bouget, Louis Hautier, Jon Sweeney, Jean-Claude Grégoire.

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
