## [Decision Letter · Decision Letter 0]

27 Nov 2023

PONE-D-23-31179Enhancing Buprestidae monitoring in Europe: trap catches increase with a fluorescent yellow colour but not with the presence of decoysPLOS ONE

Dear Dr. Kuhn,

Thank you for submitting your manuscript to PLOS ONE. After careful consideration, we feel that it has merit but does not fully meet PLOS ONE’s publication criteria as it currently stands. Therefore, we invite you to submit a revised version of the manuscript that addresses the points raised during the review process.

We look forward to receiving your revised manuscript.

Kind regards,

Francesco Porcelli, PhD

Academic Editor

PLOS ONE

Journal Requirements:

"This research was part of the AGRITRAP project (RI 2020-A337) funded by the Belgian Federal Public Service (FPS) Health, Food Chain Safety and Environment."

5. Please upload a copy of Supporting Information Figure/Table/etc. Supplements sections 1-6 which you refer to in your text on your manuscript.

Additional Editor Comments:

Dear Authors,

four reviewers examined your submission giving different opinion about the suitability for publication.

My suggestion is to apply major revision to the study, re-presenting it to a second round of revisions.

Please consider first the Reviewers 1 & 3 suggestions, being the most demanding.

Thank you very much.

Reviewers' comments:

Reviewer's Responses to Questions

**Comments to the Author**

1. Is the manuscript technically sound, and do the data support the conclusions?

Reviewer #1: Partly

Reviewer #2: Yes

Reviewer #3: No

Reviewer #4: Yes

2. Has the statistical analysis been performed appropriately and rigorously? 

Reviewer #1: No

Reviewer #2: Yes

Reviewer #3: Yes

Reviewer #4: Yes

3. Have the authors made all data underlying the findings in their manuscript fully available?

Reviewer #1: Yes

Reviewer #2: Yes

Reviewer #3: Yes

Reviewer #4: Yes

4. Is the manuscript presented in an intelligible fashion and written in standard English?

Reviewer #1: Yes

Reviewer #2: Yes

Reviewer #3: Yes

Reviewer #4: Yes

5. Review Comments to the Author

Reviewer #1: My review is presented below. I attached the supplementary excel file (with data issues presented - for details see review). Unfortunately, I had to reject the manuscript, because I found major issues related to the experimental design, data handling and statistical analyses. Relative to English - there are some minor issues related to grammar and style (as far as I could find as a non-native speaker), which are also listed in my review.

5 November 2023

Review for PLOS ONE

Manuscript entitled „Enhancing Buprestidae monitoring in Europe: trap catches increase with a fluorescent yellow colour but not with the presence of decoys”

Authors: Alexandre Kuhn, Gilles San Martin, Séverine Hasbroucq, Tim Beliën, Jochem Bonte, Christophe Bouget, Louis Hautier, Jon Sweeney, Jean-Claude Grégoire

In the manuscript, the authors presented the results of the evaluation of different trap types and colours as well as presence of decoys on catches of the jewel beetles in different types of sites.

Introduction provides sufficient information on the current state of knowledge and motivation to initiate further studies.

Unfortunately, experimental design was not well prepared and/or organized/harmonized, thus causing data analyses complicated. Most of shortcomings of the experimental design are presented only in supporting file, but not in the main body of the manuscript, thus the current description of the methods makes an impression that the studies were well developed and conducted, while, in fact, there were many issues. The experimental design have to be described in details to make sure/clear that data were collected and analysed properly.

My major concerns are:

1. The period of trap exposure and frequency of inspections presented in line 129 do not correspond to what can be found in further description, Table 1 and file with raw data. Both time of exposure and frequency of inspections varied much depending on year, sites etc. Authors wrote that they used randomised block design, in which trees (blocks) were supposed to be nested in sites, but actual design was very unbalanced. E.g., in 2021, four types of traps were tested in 13 sites. In each site, sets of four trap types were supposed to be hung in the canopy of three trees, however two (Multifunnel green and MULTz yellow) of four trap types were hung in the canopy of two trees, instead of three. In Table 1, the design looks even more complicated: in Fagus sites (two sites) – the number of Multifunnel green traps should be 4 in total (2 sites x 2 traps), while there were only 2 traps; the number of MULTz yellow traps should be also 4, but there were 5. In Quercus sites one could expect 12 traps of Multifunnel, but there were 11. Similar issues were in 2022. It also appeared that different types of traps within blocks were exposed for different periods, varying in 2021 in Fagus sites from 37 to 107 days, in Pyrus sites – from 38 to 101 days and in Quercus sites – from 91.7 to 100.5 days. Similar issues were in 2022. It is getting even worse, when one checks the file with raw data – there were very few cases when all tested trap types were exposed in the field simultaneously (in the same periods). The authors seem to neglect this and tried to standardise the sampling effort by calculating the catches per day and then multiplying by 90 days, however such approach cannot be used when different traps within a block were exposed in different periods, because flight activity of insects is not constant through the time. It is crucial to analyse data only for the period, during which all compared trap types in a block (a tree) were exposed in the field simultaneously. In the excel file attached to this review, the authors could see what data can be used (in transparent raws) or cannot be used (grey colour) for testing the effect of trap type. Data cannot be used because the number of captured beetles is missing for at least one trap type. Sites/raws in the blue colour can be skipped because of all zero values. In France, all tested trap types were exposed during the same period only in three sites (although green fan traps were hung on other tree than remaining traps) and the period was very short – from 29 June till 28 July. Only after appropriate selection of data, the authors could standardize the catches obtained in Belgium (usually covering period longer than 90 days) by calculating them for the period of 90 days. Data from France that can be included in analyses are available only from the period of 29 days, basically in July, when catches of Agrilus were already much lower than in May and June, thus standardizing them in the same way as the Belgian data would lead to great over/underestimation of catches, depending on Agrilus species.

2. Data from two years cannot be combined in one data set for the analysis, because there were two different trap sets tested, therefore bottle traps cannot be compared with fan traps. Data from two years have to be analysed separately.

3. Data from traps of the same type but with and without decoy cannot be summed (line 251), but have to be averaged – you compare trap type, not doubled trap type. The issue of trap size have to be included in the discussion. If you have mixed data (not divided depending on the presence of decoy) and you want to include them data into analyses (if the effect of decoy appeared not significant), you have to divide the results by 2.

4. Data for testing the effect of decoy on trap catches, effect of trap type on the number of species and probability of detecting each selected insect species have to be checked and carefully selected for analyses considering conditions mentioned in points 1-3.

5. The number of replications is not sufficient for creating models with many factors and variants/factor, particularly for models including insect species (with 7 species), e.g. trap_type*decoy*species and with more factors and interactions. I would suggest to analyse data for each selected insect species and year separately, using appropriate data (see point 1). Such approach will impose the use of data only from oak stands in the analyses. Testing the effect of any factor on combined catches of all insect species seems unreasonable – in addition to concerns mentioned above, the results will show rather the patterns characterising the most abundant insect species, while the effects on less abundant species will be hidden.

6. I would suggest to use data without log-transformation (pooled number of specimens of a certain species captured during appropriate period – see point 1, then standardized and rounded to integers) and GLMM with either Poisson or negative binomial distribution of residuals for testing the effect of trap type and sex or decoy and sex. When such data are used, you don’t need to account for differences in sampling effort as long as you use results for the period (dates) when all trap types in a block (on a tree) were exposed simultaneously. It seems to me that you incorrectly specified the GLMM models (lines 226-229). By the way, if you use notation of random factors in the following way: (1|SiteID) + (1|TreeID), it would be useful to make a note in M&M that Tree ID was unique for each site.

7. I am not sure why tree species was included in some models.

8. Probability of detection of different species by different trap types is worth to test probably only for most abundant species. The authors used model (Suppl. Material)

glmer(Presence ~ Trap_Type*Species + (1|SiteID) + (1|TreeID),family = binomial),

but it seems to be written in a wrong way. In binomial model, the dependent variable has to be presented as the number of traps with species presence divided by the total number of traps and the model has to include weights = total number of traps, i.e. the model (for each insect species separately) would have to look as following:

glmer(number of traps with species presence/total number of traps ~ Trap_Type + (1|SiteID) + (1|TreeID), weights= total number of traps, data=data, family= binomial).

9. Lines 392-393 – I am not sure why you analysed correlation between the number of species and the number of insects – there is no information on this point in M&M.

10. Results and Discussion will have to be rewritten after the proper analyses using carefully selected data are done.

Besides major issues related to the experimental design and data analyses mentioned above, there are also other issues.

11. In M&M, it is necessary to provide more information on: 1) forests/orchards in each site – age, density, species composition and stand structure (layers, presence of undergrowth/understory) (shorter description in the main body of the manuscript and more details – in Suppl.Info), 2) distance from a trunk where traps were hung – was it comparable for each trap hung in a crown, 3) approximate height of trap location above the ground and how traps were hung in the upper canopy (were all traps in each site hung there), 4) location of trees with traps –interior or edge of a forest/orchard. All this information could be important for understanding/interpreting the results, e.g. differences in species composition and abundance of buprestids in Belgium and France.

12. It is necessary to explain why decoys of A. planipennis (alien species not present in Europe, except Ukraine and western part of Russia) was used for testing the effect of decoy on catches of Agrilus species native to Europe, particularly if one could expect species-specific response.

13. Latin names of insects have to be followed by their authors.

14. Why the authors focused on buprestids in broad-leaved species? What about the species in conifer forests, e.g. Phaenops cyanea? There was not a word in introduction or in discussion about those species, while the species that are not green might provide some insights into the relationships between colour of trap and the colour of insect and/or host tree.

15. It would be more logical to describe all analyses in one chapter called e.g. Statistical analyses, starting from description of data preparation, then what was tested and how (type of models, model structure, data selection, etc.) and then programs/scripts used for analyses.

16. In Discussion, I would make a point about the importance of colour and size of different buprestid species on catches in traps of certain colours and with decoys of certain size/colour. This might be useful in explaining low catches of A. sinuatus.

17. I wonder why neither in Results, not in Discussion no attention was paid to close match of main peaks of reflectance in fan-trap yellow and multifunnel green traps that could explain their similarity in catches. I would suggest to move the description of the results of reflectance measurements from M&M to Results and add one figure in the main manuscript to show this match.

18. Lines 477–481 – is it about the results presented in the manuscript or about the results discussed earlier (ref. 46)?

19. Files in Supporting Materials, particularly the way how raw data are presented, are not really helpful in understanding what data were used for each analysis. There are much more data than are directly related to the manuscript. Closer look at the raw data showed that there are major issues with experimental design and proper selection of data for analyses is crucial.

20. Figures – the authors should avoid presenting non-significant results in Figures, while all the significant results should be well demonstrated, either in figures or in tables.

21. I could not find the information about the contribution of each author.

22. Explain all abbreviations, e.g. PTFE, GLMM etc. at the first place you introduce them. It would be better to avoid using NB for specifying some information.

There are some grammatical and style issues:

Line 30 – write “two-week basis” instead of “2-week basis”; numbers 1-9 usually are written in words.

Line 60 – delete dot after “traps”

Line 138 – MULTz was yellow, not yellow-green

Line 144 and 182 – “with PTFE for dry surface” – I am not sure what it means,

Line 143 and 181 – “attached them by back-to-back couples” - I am not sure what it means

Line 145 – “in the upper canopy” – pictures in Fig. 1 suggest that the traps were not hung in upper canopy – please, provide clear information on where and how the traps were hung (see point 11).

Line 151 – please write more clearly – was each trap type on a different tree or there were some combinations? Data in supplementary file shows that all four trap types were tested in 4 of 5 sites and only in the period from 9 August till 16 September.

Line 161 – I presume that the number of samples means the number of inspections of each trap. Am I write?

Line 179 – “the” first and second layer

Line 210–212 – need corrections – a) residual plots are not used for checking “the conditions of application of the model”; b) “the choice of transformations” – what does it mean?, 3) “(61, 62); see Supplements)” – do not help in understanding what exactly you did, and please correct parentheses.

Line 215–216 – please correct style ¬ a) treatment doesn’t catch insects, 2) “specimen of the particular response variable being analysed” – what does it mean?

Line 529–532 – 1) structure of sentence has to be corrected, 2) colour closer to – “closer” is not a correct term in this case; you could use “similar”, “corresponds to”, “reflects” or something similar.

Reviewer #2: The manuscript is well written. It is about an extensive series of 36 field experiments in different vegetations to test trapping devices suitable for jewel beetle monitoring. Further, the manuscript includes developing a novel trap device for jewel beetles. Lab measurements and a sufficient review of the up-to-date literature also support the work. Their result helps the understanding of the state of the art of jewel beetle monitoring devices and prove statistical conclusions that I myself had a gut feeling of but had no sufficient data set to prove like the abundance of jewel beetles in different habitats, the importance of the tree on which the monitoring traps are placed, which demonstrates how important trap rotation is within experiments with jewel beetles. Altogether the present work adds to our knowledge in the field of trapping Agrilus jewel beetles.

Reviewer #3: Overview: This work uses a few different trap designs to test for their ability to capture Agrilus beetles on different tree types in European forests and orchards. The authors collect useful data that indicates how certain species may be caught using these different trap designs. While the data collected is interesting and useful, the presentation of the data falls far short of presenting the information to the reader in an understandable and relevant form.

First, there is far too much focus on visual decoys, with little understanding evident as to how they might work in trapping systems. The decoy trap studies referenced by previous authors, make it clear that decoys only work in branch-based traps where the traps consist of small leaf-sized surfaces that are wrapped around terminal branches and placed directly within the live foliage of the trees. Those previous studies showing that decoys can work in these situations, as well as other studies have always found that decoys do not work well in larger stand-alone traps such as funnel traps or prism traps. Thus, the lack of an effect of decoys in this study as the authors designed the experiment is entirely expected and not worthy of such in depth discussion as they dedicate to the topic.

It would be better to simply mentioned that the decoys were used on some of the traps and present the statistical support that they did not have a significant effect, without dedicating an entire Figure to this point (Fig 3)

It would be a better use of the data, to more present how the different tree types affected the species of insect trapped. The authors do mention that tree type did have a statistically significant effect on total numbers caught. However, it would be very interesting to see which species were caught on the different tree types. The authors focus instead on country rather than tree species to describe the species as they did in figure 2. However, a figure similar to figure 2 showing the different tree species and the distribution of species caught would be more interesting.

It would be too much to present in figure form, but a Table listing each combination of tree and trap type with the corresponding numbers of each species caught would also be interesting and helpful for understanding which traps might be best to use for each species.

The effect of color is only tested for one trap type (the fan type) This is helpful as well, but as the authors mention, the yellow is better for most species, but not all, so it should not be implied strongly that the yellow traps are optimal. Agrilus biguttatus is a much more economically destructive pest than many of the others described, so it’s preference for green funnel traps may be highly important to a monitoring or surveillance program.

More specific comments on particular sections:

Title: Very specific conclusions are presented in the title, which are not strongly supported by the data and/or based on a less than optimal design. A broader title without such specific conclusions would be more appropriate, such as “A comparison of trap designs for capturing Agrilus species in multiple forest and orchard settings in Europe”

Ln 107: Add here that that decoy-baited traps have only been shown to be effective when designed to be placed on branches within the living foliage. 21, 46 and also Domingue et al., Journal of Pest Science 88, 267-279

Ln 108-116: This description of the need for traps to be placed high in the canopy and easily accessible also fits in to the issue of decoys. Because decoys only work in branch traps that can be reached and serviced easily from the ground on low lying branches, they are less convenient. They also only work in sticky designs. Therefore, it would be better to justify your study examining the role of decoys on these multifunnel and fan traps to be sure they are not effective in these situations as the previous literature would also predict that they would not be.

Figure 1 caption: what type of site, with what tree species is being pictured here in each panel?

Figure 2: Does this include pool all the tree species? It should be explicitly mentioned in the caption.

Figure 4: It is not necessary to include the word “mixed here” You had demonstrated in figure 3 that pooling the traps with decoys is not an issue. You also do not use the designation in Figure 5. It just confuses the reader to have it here.

Ln 441: It would be worth mentioning that branch trap designs might be worth investigating in orchard settings such as this (ref 21&46) if these other trap designs are failing.

Ln 451-481: the authors spend a disproportionate amount of time discussing previous decoy studies and questioning the statistical validity of these studies and minor methodological issues. They are neglecting to discuss the much more important consideration that the trap design used for branch traps were much different than any of the trap designs used in this study.

The branch traps place the decoys directly in among the leaves of the tree, which is likely the crucial factor in providing enough visual and olfactory stimuli to allow the decoys to work. In a stand-alone trap placed in trees, hanging far from the branches it seems likely that the mating behavior exhibited by the males in response to the decoys simply cannot be reproduced.

The authors also question the validity of the statistics of the branch trap studies as a result of the failure to account for experiment-wise error. However, the main finding was based on a pooling of all species, which was highly significant. Questioning the significance of decoys for each species here seems to be wondering well outside the scope of this study and what conclusions can be made about decoys from these authors’ results.

Ln 482-521: The authors make an interesting point about the economic and usability advantages of fan traps. Some quantification of the costs of each trap type should be included.

Reviewer #4: This manuscript reports on field trapping experiments in Belgium and France to compare efficacy of different trap types and colors with or without dead Agrilus planipennis decoys for capturing buprestid species including known pests of oak, beech, and pear trees. Overall the paper is well-written and it provides new information that will improve monitoring surveys for buprestid beetles that are among the most damaging forest pests and highly successful invaders. The experiments were well designed and were replicated across multiple sites per year in Belgium and France, the results are clearly presented and the discussion and conclusions are supported by the data. I don’t have any major concerns with the paper and believe it is suitable for publication with some minor revisions.

1. The abstract (lines 30-31) states that over 2 years, 382 traps were deployed across 46 sites in Belgium and France. However, in the methods it states that 4 trap designs were compared at 13 Belgium sites in 2021 (line 132) with 3 replicates (line 145) per site (13*4*3=156 traps) and 6 trap designs were compared at 23 sites located in Belgium or France (line 166) in 2022 with 2 replicates (line 188) per site (6*23*2=276) for a total of 36 sites (13+23) and 432 (156+276) traps – which does not agree with the totals in the abstract.

2. It would be helpful to include a little more detail in the methods. Were traps baited with any lures? How were samples collected from the collecting pots – were the contents strained and placed in bags and how were samples stored until counted? Since all trap types were hung in the same tree for each replicate, could there have been any cross attraction or interference? Were the traps hung in different aspects/cardinal directions of the tree canopy? Were trap types rotated around the tree canopy to different aspect positions between collection periods? Canopy position/aspect and condition (open/edge, proximity to adjacent trees) can impact attraction of different species.

3. Spectral data for the traps were measured (line 191-203) but are not discussed in the results or discussion. How does the spectral reflectance of the different traps compare to electro-retinogram studies (if available) or leaf color?

4. Why were trap catches standardized per 90 day period instead of the 2 week sampling period between collections?

5. Consider including statistics for the statement on line 266-267 that year was not included in the model because it seemed to have little effect. The statistics for the non-significant effect of year are provided at line 339.

6. What multiple comparison procedure was used to compare all pairwise comparisons (line 270).

7. Any idea why yellow/green color might be less attractive to A sinuatus in pear orchards, given that pears are yellowish green in color? (line 441)

8. Does A. sinuatus fly at lower heights in pear orchards that typically have shorter trees than forests? (line 446)

9. What about testing decoys of the target species rather than A. planipennis? (line 451)

10. Recommend changing “profitable” to “cost effective” (line 518) profitable implies the traps will generate income (make money)

11. Was there a balanced sex ratio for all species (line 522)

12. Recommend changing “pay off” to “increase detection and captures” or “optimize detection and captures” (line 539) - again "pay off" suggests monetary gain.

13. Consider dropping the word “interestingly” in several places - it is subjective.

14. Consider changing passive to active voice in several places (e.g, change “for the monitoring of buprestids” to “for monitoring buprestids”).

15. Drop “To summarize” at the beginning of the conclusions section. It is redundant since a conclusion is clearly a summary.

16. Could break up a few long paragrphs in the discussion that cover multiple topics. For instance the paragraph starting at line 425 could be broken at line 438 to separate discussion of the impact of tree and site characteristics form the impact of color and height.

6. PLOS authors have the option to publish the peer review history of their article (what does this mean?). If published, this will include your full peer review and any attached files.

Reviewer #1: No

Reviewer #2: No

Reviewer #3: No

Reviewer #4: No

---

## [Author Response · Author response to Decision Letter 0]

21 Dec 2023

We have addressed all of the reviewers' comments in the Response to Reviewers file.

---

## [Decision Letter · Decision Letter 1]

7 May 2024

PONE-D-23-31179R1Enhancing Buprestidae monitoring in Europe: trap catches increase with a fluorescent yellow colour but not with the presence of decoysPLOS ONE

Dear Dr. Kuhn,

Thank you for submitting your manuscript to PLOS ONE. After careful consideration, we feel that it has merit but does not fully meet PLOS ONE’s publication criteria as it currently stands. Therefore, we invite you to submit a revised version of the manuscript that addresses the points raised during the review process.

We look forward to receiving your revised manuscript.

Kind regards,

Ramzi Mansour

Academic Editor

PLOS ONE

Additional Editor Comments:

Please add the authorship as well as both the order and family between brackets to each insect species mentioned for the first time in the text (example in Line 69 (Introduction): Agrilus planipennis Fairmaire (Coleoptera: Buprestidae)). Additionally, in Line 70, please replace "Agrilus"  with  "A."  and in Line 71, replace "Coraebus"  with  "C.".

Reviewers' comments:

Reviewer's Responses to Questions

**Comments to the Author**

1. If the authors have adequately addressed your comments raised in a previous round of review and you feel that this manuscript is now acceptable for publication, you may indicate that here to bypass the “Comments to the Author” section, enter your conflict of interest statement in the “Confidential to Editor” section, and submit your "Accept" recommendation.

Reviewer #1: (No Response)

Reviewer #2: All comments have been addressed

Reviewer #3: All comments have been addressed

Reviewer #4: All comments have been addressed

2. Is the manuscript technically sound, and do the data support the conclusions?

Reviewer #1: Partly

Reviewer #2: Yes

Reviewer #3: Yes

Reviewer #4: Yes

3. Has the statistical analysis been performed appropriately and rigorously? 

Reviewer #1: No

Reviewer #2: Yes

Reviewer #3: Yes

Reviewer #4: Yes

4. Have the authors made all data underlying the findings in their manuscript fully available?

Reviewer #1: Yes

Reviewer #2: Yes

Reviewer #3: Yes

Reviewer #4: Yes

5. Is the manuscript presented in an intelligible fashion and written in standard English?

Reviewer #1: Yes

Reviewer #2: Yes

Reviewer #3: Yes

Reviewer #4: Yes

6. Review Comments to the Author

Reviewer #1: 

Dear Authors,

Unfortunately, you haven't addressed most of my comments sufficiently, particularly the most crucial points, therefore I sustain them and will not be able to accept the manuscript as long as data are not managed, analyzed and presented correctly. See attached document for detailed comments.

Reviewer #2: 

Congratulations on the manuscript, which is well-written and suitable for publication. The results help the understanding of our present limits regarding jewel beetle monitoring devices proven with statistics. Often discussed opinions about jewel beetle monitoring become proven facts with the support of the present work.

Reviewer #3: 

No additional comments needed. The meticulous attention to all the comments have greatly improved the paper.

Reviewer #4: 

The authors have done a great job addressing all of my comments on the original version. I believe the manuscript is now suitable for publication

7. PLOS authors have the option to publish the peer review history of their article (what does this mean?). If published, this will include your full peer review and any attached files.

Reviewer #1: No

Reviewer #2: No

Reviewer #3: No

Reviewer #4: **Yes: **Therese Poland

---

## [Author Response · Author response to Decision Letter 1]

3 Jul 2024

Response to Reviewer #1:

Please see our response to the editor.

Response to Reviewer #2: 

We thank Reviewer #2 for the very positive feedback on our study.

Response to Reviewer #3:

We are glad that our revised version of the manuscript satisfies Reviewer #3 and thank him/her for the time spent on it.

Response to Reviewer #4:

 We are pleased that our revisions addressed all of Reviewer #4's concerns, especially given her expertise in this area, and thank her for helping us improve our manuscript.

---

## [Editor Report · Decision Letter 2]

4 Jul 2024

Enhancing Buprestidae monitoring in Europe: trap catches increase with a fluorescent yellow colour but not with the presence of decoys

PONE-D-23-31179R2

Dear Dr. Kuhn,

We’re pleased to inform you that your manuscript has been judged scientifically suitable for publication and will be formally accepted for publication once it meets all outstanding technical requirements.

Kind regards,

Ramzi Mansour

Academic Editor

PLOS ONE

---

## [Editor Report · Acceptance letter]

9 Jul 2024

PONE-D-23-31179R2 

PLOS ONE

Dear Dr. Kuhn, 

I'm pleased to inform you that your manuscript has been deemed suitable for publication in PLOS ONE. Congratulations! Your manuscript is now being handed over to our production team.

Kind regards, 

on behalf of

Dr. Ramzi Mansour 

Academic Editor

PLOS ONE